# Metabolomic Profiles in Starved Light Breed Horses during the Refeeding Process

**DOI:** 10.3390/ani12192527

**Published:** 2022-09-21

**Authors:** Sawyer C. Main, Lindsay P. Brown, Kelly R. Melvin, Shawn R. Campagna, Brynn H. Voy, Hector F. Castro, Lewrell G. Strickland, Melissa T. Hines, Robert D. Jacobs, Mary E. Gordon, Jennie L. Z. Ivey

**Affiliations:** 1Department of Animal Science, University of Tennessee, 2506 River Drive, Knoxville, TN 37996, USA; 2Department of Chemistry, University of Tennessee, 1420 Circle Drive, Knoxville, TN 37996, USA; 3Biological and Small Molecule Mass Spectrometry Core, University of Tennessee, 1416 Circle Drive, Knoxville, TN 37996, USA; 4Department of Nutrition, University of Tennessee, 1215 W. Cumberland Ave., Knoxville, TN 37996, USA; 5College of Veterinary Medicine, University of Tennessee, 2407 River Drive, Knoxville, TN 37996, USA; 6Purina Animal Nutrition, 100 Danforth Drive, Gray Summit, MO 63039, USA

**Keywords:** equine, malnourished, emaciation, body condition score, metabolomics, refeeding, rehabilitation, high resolution mass spectrometry

## Abstract

**Simple Summary:**

Unwanted horses and other equids continue to be a global issue throughout which the prevalence of malnourished equids persists as a welfare concern. Nutritional studies evaluating the refeeding or rehabilitation process are limited, and little data exists to explain how metabolic function changes during refeeding. Therefore, the objective of this study was to assess changes in metabolite profile (metabolome) of emaciated horses throughout rehabilitation via refeeding. Horses were fed specific diets over the rehabilitation period, and blood samples were collected and analyzed throughout. Comparison of equine within a malnourished state and equine during the refeeding process showed decreases in potentially toxic chemical compounds related to liver, kidney, and muscle function as well as shifts related to changing energy demands. This study concludes that the refeeding and rehabilitation process results in an overall change in the equine metabolome.

**Abstract:**

The large population of emaciated horses continues to be an issue troubling the equine industry. However, little is known regarding the collection of equine metabolites (metabolome) during a malnourished state and the changes that occur throughout nutritional rehabilitation. In this study, ten emaciated horses underwent a refeeding process, during which blood samples were collected for a blood chemistry panel and metabolomics analysis via ultrahigh performance liquid chromatography–high resolution mass spectrometry (UHPLC-HRMS). Significant differences among blood chemistry analytes and metabolite abundance during the critical care period (CCP; Days 1–10 of rehabilitation) and the recovery period (RP; the remainder of the rehabilitation process) were observed. Potentially toxic compounds, analytes related to liver, kidney, and muscle function, as well as energy-related metabolites were altered during the refeeding process. The combination of blood chemistry and metabolomics analyses on starved equine during rehabilitation provide vital biological insight and evidence that the refeeding process has a significant impact on the equine metabolome.

## 1. Introduction

The equine industry continues to be plagued with a significant number of “unwanted” horses and other equids [1] with an estimated 138,000 equids deemed unwanted in the United States alone each year [2]. For various reasons (including lack of nutritional knowledge by the owner, misalignment of expectation to actual cost of equine ownership, and in some cases, criminal neglect), many unwanted equids can become emaciated [1]. Despite the commonality of emaciated horses, there has been minimal research dedicated to the evaluation of their nutritional recovery and metabolic repair. The literature describes some studies on rehabilitating or refeeding the starved horse, yet these studies span only 10 to 15 days, while the average refeeding period lasts for several months [3,4].

Malnutrition is diagnosed as a lack of nutrition due to inadequate or unbalanced intake of nutrients, or their impaired assimilation or utilization [5]. Severe malnutrition or starvation yields proteolytic and lipolytic responses that can result in negative physiological issues including refeeding syndrome, cachexia, serous atrophy, muscle contractions due to low calcium stores, and heart arrythmia [6]. The Henneke body condition scoring system is an important tool for the rapid, qualitative assessment of nutritional status; however, this system cannot assess the underlying factors associated with respective changes in body composition, metabolism, and overall health [7]. Because malnutrition is a complex condition that is influenced by an array of factors, it is often difficult to determine and label the true state of malnutrition in equids, for which starvation is typically classified using a low body condition score (BCS; <3 on a 1–9 scale) [1,7]. Elements influencing nutritional status including genetics, background, predisposition to disease, and dietary history are often unidentified within the unwanted and starved horse populations. Thus, a consistent evaluation method coupled with an understanding of changes in metabolism is important to understanding emaciated phenotypes and, if possible, a recovery to a healthy body weight and condition.

A more quantitative approach to evaluating a biological system is that of metabolomics, the global measurement of the suite of small molecule metabolites that constitute an organism’s metabolome. The metabolome is made up of numerous small molecules that are derived from or utilized by all known metabolic pathways. Metabolomics provides insight into an organism’s physiological state (i.e., phenotype) and, thus, can serve as a quantitative, global measurement to supplement other qualitative approaches, such as the BCS system [7,8]. Investigating the metabolic profile of emaciated equids could provide information as to which metabolic pathways are affected during a state of malnutrition [9].

At the present time, little research has been conducted in the field of equine metabolomics regarding nutritional rehabilitation. Current equine studies mainly focused on healthy thoroughbreds, pregnant mares, exercise, and doping [10,11,12,13]. Metabolomics research conducted in malnourished pigs and humans with cachexia can serve as a foundational comparison for emaciated equids, but species-specific knowledge through dietary intervention periods is warranted [9,14,15]. Despite the vast array of information available pertaining to the increasingly popular field of metabolomics, no studies are dedicated to the malnourished horse or the refeeding and rehabilitation process over the nutritional recovery period. Research pertaining to the equine metabolome throughout stages of malnutrition and the refeeding process could provide insight into specific nutritional requirements and allow owners to make dietary adjustments accordingly. Thus, this study focused on evaluating the metabolome of various malnourished horses during the refeeding process to determine metabolic differences between the starved and rehabilitated horse during the refeeding process.

## 2. Materials and Methods

### 2.1. Equine Enrollment

Ten light breed horses aged 19 ± 3 years were enrolled into the study over a three-month period (Table 1). Horses for the study were obtained from three outlets: (1) in cooperation with the Tennessee Department of Agriculture (TDA) and the Tennessee State Veterinarian by owner surrender as a result of neglect cases, (2) purchased from public auction, and (3) private owner surrender. Enrollment criteria included: (1) BCS of 2 or less [7], (2) light breed mares or geldings, and (3) apparent good health, aside from malnourishment, as defined by temperature, pulse and respiration rates within normal limits, and no evidence of coughing or nasal discharge at the time of enrollment. Due to the nature of horse acquisition and enrollment, previous diet history on each horse could not be acquired. All study procedures were approved by the University of Tennessee Institutional Animal Care and Use Committee (Protocol #2811-1220).

### 2.2. Study Timeline and Sample Collection

The first ten days of the study were labeled as the critical care period (CCP). During this period, horses were housed and maintained at the University of Tennessee College of Veterinary Medicine (UTCVM; Knoxville, TN, USA) where their initial health status was evaluated by a veterinarian and a Coggins test was performed. Blood samples were collected from the jugular vein on the day of intake and every other day thereafter. Weekly, horses’ BCS were assessed by two individual, trained reviewers [1]. Vital signs including temperature, pulse, respiration, capillary refill time, and skin tent were assessed every 12 h during the CCP to monitor horses for disease onset and overall health status in compliance with UTCVM policy. If horses displayed signs of potential infection, a respiratory swab was performed to check for Equine Herpes Virus (EHV), Strangles, and Influenza. If horses tested positive for any of these diseases, attending veterinarians worked in conjunction to provide care and treatment while maintaining refeeding protocols. Body weight (BW) was assessed daily using a commercially available equine scale (Optima, Rancho Cucamonga, CA, USA).

The remainder of the study duration (days 10–165) was designated as the recovery period (RP). During the RP, horses were housed and maintained at either UTCVM, the University of Tennessee Veterinary Research and Education Center (VREC; Knoxville, TN, USA), or at Middle Tennessee Research and Education Center (MTREC; Spring Hill, TN, USA), based on housing availability. Horses were always moved as a cohort, and at least six days of acclimatization was allowed before the next sampling period. Before transportation, all horses tested negative for infectious diseases.

Weekly BW measurements were taken throughout the duration of the study. Every two weeks, BCS was evaluated by two independent trained reviewers for each horse and jugular venipuncture blood samples were collected at each BCS increase. Fecal samples and respective fecal float and centrifugation tests were performed on Day 14 and horses were dewormed based on fecal test results. Horses were determined to progress to the next BCS when all reviewers agreed that the next BCS was reached, and a weight gain of 16–20 kg from the previous BCS was achieved [16,17]. If horses reached a BCS of 4 before the end of the rehabilitation period, final measurements and samples were taken prior to study disenrollment. Plasma blood samples were maintained on ice until centrifugation at 3000× *g* for 10 min. Serum blood samples were allowed to clot overnight and then centrifuged at the same speed and duration. All aliquots were stored at −80 °C until analysis.

### 2.3. Diet Formulation

Upon enrollment, each horse was randomly assigned one of two diet groups for the duration of the study: control (C) diet which consisted of timothy hay and a commercially available vitamin mineral product (Purina Free Balance) where diets were formulated to meet DE requirements from forage alone; and senior (S) diet, in which 50% of DE requirements were provided from a commercially available complete feed, and the remaining 50% provided by timothy hay (Table 2). Diets were formulated to be isocaloric and meet daily National Research Council (NRC) nutrient requirements for mature horses weighing under 600 kg.

The refeeding diet protocol was established based on previously published methods [3,18]. During the CCP, from day of intake until Day 3, horses were fed 50% of daily digestible energy (DE) requirements split into six feedings. On Days 4 and 5, feeding amount increased to provide 75% of daily DE requirement over six feedings. On Days 6 to 10, horses were fed 100% of daily DE requirement over three different feedings.

Diets were formulated to provide adequate DE for 0.45 kg of BW gain per day. Initially, DE was increased to 125% of daily DE requirements based on weekly BW measurements. If horses did not gain weight at an appropriate rate for two consecutive BW measurements, DE was increased by 10%, to 135 or 145%, respectively.

### 2.4. Blood Chemistry Analysis

Plasma and serum chemistry quantitative values (Table 3) were determined via photometric absorbance values utilizing a COBAS C311 analyzer under guidelines determined by the manufacturer and utilizing approved reagent and standards (Roche, Switzerland).

### 2.5. Metabolomics Extractions

The metabolomics extraction procedure was adapted from Rabinowitz and Kimball [19]. Briefly, 100 µL of plasma was mixed with 1.3 mL of chilled extraction solvent (4:4:2 acetonitrile:methanol:water with 0.1% formic acid) in an Eppendorf tube and allowed to extract for 20 min at −20 °C. Samples were centrifuged and the supernatant was pooled into a separate glass vial. To the solution remaining in the Eppendorf tube, 200 µL of extraction solvent was added, the mixture was vortexed, and allowed to extract for 20 min at −20 °C. Samples were then centrifuged, and the supernatant was pooled into the glass vial with the supernatant from the first extraction. Pooled samples were dried under nitrogen and resuspended in 300 µL of water prior to mass spectrometric analysis.

### 2.6. Ultrahigh Performance Liquid Chromatography Mass Spectrometry High Resolution Mass Spectrometry (UHPLC-HRMS)

The metabolomics analyses were conducted at the Biological and Small Molecule Mass Spectrometry Core at the University of Tennessee Knoxville (Knoxville, TN, USA). The sample extractions, chromatographic separations, and mass spectral analyses were performed according to an established method using UHPLC-HRMS [20]. The collected metabolomes were stored at 4 °C in an UltiMate 3000 RS autosampler (Dionex, Sunnyvale, CA, USA) until analyses were performed. All chromatographic solvents used were HPLC grade (Fisher Scientific, Hampton, NH, USA). Reverse phase separations were carried out using a Synergi 2.6 µm Hydro RP column (100 mm × 2.1 mm, 100 Å; Phenomenex, Torrance, CA, USA) and an UltiMate 3000 pump (Dionex, Sunnyvale, CA, USA). The chromatography utilized a 25 min binary gradient elution using a 97:3 water:methanol mixture with added 10 mM tributylamine and 15 mM acetic acid (mobile phase A) and 100% methanol (mobile phase B). The flow was set to 0.200 mL/min, and the gradient was as follows: 20% B at 5 min, 55% B at 13 min, 95% B at 15.5 min, 0% B from 19 to 25 min. The separated metabolites were ionized via negative mode electrospray ionization prior to analysis on an Exactive Plus Orbitrap MS (Thermo Scientific, San Jose, CA, USA). The resolution of the mass spectrometer was set to 140,000 with an automatic gain control setting of 5 × 10^5^ and injection time of 100 milliseconds. The sheath, auxiliary, and sweep gases were set to 25, 8, and 3 arbitrary units, the spray voltage was 3.5 kV, and the capillary temperature was 320 °C.

### 2.7. Data Analysis

Descriptive statistics for refeeding parameters were assessed using STATA SE (version 16.1, StataCorp, College Station, TX, USA).

To evaluate the known metabolites, raw data collected by the instrument were converted to .mzML files via MSConvert (ProteoWizard). Features were then picked using an in-house library in EL-MAVEN based on exact mass (5 ppm) and retention time (±0.5 min). Unidentified features were evaluated via Metaboanalyst 5.0 [21]. Spectra in the form of .mzML files were compressed, uploaded to Metaboanalyst, and processed using the Auto-optimized feature in the LC-MS Spectra Processing module. As some batch effects were observed, features that were found to significantly vary (*p* value < 0.05) between batches based on an analysis of variance (ANOVA) were removed to lessen the chance of false statistical assumptions; this resulted in analysis of 26 known metabolites and 459 unidentified features. Heatmaps were generated using R with a custom script. Known and unidentified features were uploaded to Metaboanalyst for statistical analyses and visualization using the following parameters: features with missing values (<25%) replaced by one-fifth of the minimum positive value, normalized by sum, and log transformed. All other statistical analyses (e.g., partial least squares discriminant analyses or PLS-DAs) were performed using Metaboanalyst. Venn Diagrams were created using an in-house Python script.

Equine bloodwork was statistically evaluated in Metaboanalyst by performing 2-sample unpaired, equal group variance *t*-tests between various groupings (CCP vs. RP, diet, sex, and age group) using the same normalization and transformation procedure as metabolomics data. Box plots were generated using an in-house Python script.

Statistical conclusions based on *p* values were categorized as follows: *p* values of ≤0.01, ≤0.05, and ≤0.1 reflect statistical assumptions made with confidence intervals of 99%, 95%, and 90%, respectively.

## 3. Results

### 3.1. Equine Refeeding Outcomes

Originally, ten equids with a BCS of 2 or less were enrolled into the project; however, due to health circumstances, four horses were not able to complete the refeeding process. Specifically, the following situations occurred for each horse removed from the project (termed “Did not complete” in Table 4): (1) impaction colic resulting in euthanasia on Day 6 of the CCP; (2) positive respiratory swab for Equine Herpes Virus 1, wild type (EHV-1), *Streptococcus equi*, and influenza type A virus, resulting in euthanasia on Day 8 of the CCP; (3) positive respiratory swab for EHV-1, resulting in euthanasia on Day 1 within the RP; and (4) positive respiratory swab for influenza virus type A resulting in subsequent pneumonia, and due to lack of body weight gain was disenrolled on Day 73 of the RP. The remaining six horses (C diet *n* = 3 and S diet *n* = 3) were refed successfully to either a BCS of 3.5 or 4 (termed “Completed” in Table 4).

Initial weight at enrollment was 368 ± 58 kg (mean ± standard deviation; *n* = 10) and final weight after refeeding was 426 ± 54 kg (*n* = 6). Of the six horses that completed the project, the entire refeeding process from enrollment to successful disenrollment lasted 129 ± 28 days.

### 3.2. Analysis of Blood Chemistry Panels

Box plots were created to display differences in blood chemistry analyte levels between the CCP and RP (Figure 1). Results indicated statistical significance in analytes present in blood plasma samples (Appendix A). LDH, ALB, AST, and CREAT levels were all statistically increased during the RP compared to the CCP. Evaluation of blood serum corroborated a significant change in LDH, ALB, and CREAT (Appendix A). Non-esterified fatty acids were also shown to significantly decrease in serum during the RP. Statistical differences were also detected when comparing diet, sex, and age group in blood plasma and serum samples (Appendix A).

### 3.3. Fold Change Analysis of Metabolites Detected in Equine Subjects during the CCP and RPs

Fold changes (i.e., comparisons of metabolite abundance between sample types) were calculated for metabolites present in blood plasma. Of the metabolites detected, those classified as amino acids (amino acids, precursors, derivatives, and metabolism byproducts) showed the most perturbations (Figure 2). Overall, amino acid related metabolites such as glutamine, methionine, phenylalanine, alanine/sarcosine, and creatine showed statistically significant decreases during the RP compared to the CCP; however, *N*-acetylornithine showed a significant increase in abundance during the RP. Additionally, metabolites such as uridine, allantoin, and uric acid showed significant decreases. With the exception of *N*-acetylornithine, metabolites showing significant fold changes exhibited an overall decreased abundance during the RP compared to the CCP (Appendix A).

### 3.4. Differences of the Metabolic Profiles of Equine Subjects in the CCP vs. the RP

PLS-DA multivariate analyses allow complex data (i.e., metabolomics data) to be reduced to a simple three-dimensional scores plot, which clusters data based on variability between the samples [17,22]. Using these plots, one can generate hypotheses as to potential differences in metabolomic profiles between groupings, as well as investigate the metabolites that drive those differences. A PLS-DA was performed on the metabolomics data (known metabolites and unidentified features) to investigate potential differences in metabolomic profiles between equine in the CCP vs. the RP. Comparison of features unique to these groups resulted in a clear separation (Figure 3). As the known metabolites represent only a small portion of the data presented here, unidentified features were included in this analysis as these features provide insight into metabolic processes that have not yet been well described and contribute to a global view of all characteristic data points in a sample, serving as an important component of metabolic profile. Data from these analyses, combined with statistical fold change analysis, suggest that differences in the metabolome can be observed for equine subjects in the CCP vs. those in the RP undergoing the refeeding process.

### 3.5. Effects of Subject-Specific Attributes on the Metabolome

Metabolic profiles (including known metabolites and unidentified features) were compared using the variety of metadata provided within the study: diet (C vs. S), sex (mare vs. gelding), and age group (below 20 years old vs. 20 years old and above). Analysis on breed specifications were not conducted due to low replicates (e.g., only one equine subject was of the Standardbred breed). PLS-DAs comparing diet, sex, and age group showed distinct groupings (Figure 4). Features were evaluated using a *t*-test and yielded 127, 157, and 40 statistically significant compounds (*p* value ≤ 0.1) for diet, sex, and age comparisons, respectively (Appendix A).

Components that drive separation in the PLS-DA are characterized by a higher variable importance projection (VIP) score (i.e., VIP scores greater than 1; Appendix A). A Venn Diagram was used (Figure 4) to gauge feature-based similarities and differences between subject-specific attributes. Overall, 33 features (including known metabolites and unidentified features) were common to all three groups evaluated, suggesting these features are common across the equine used in this study. Features that were unique to each group were much higher for sex and diet (22 and 28, respectively) than that of age group, which possessed eight unique features.

## 4. Discussion

### 4.1. Animal Origin and Source

Large animal nutrition studies create unique opportunities and challenges due to management, availability, and cost for such trials. Within our study, emaciated animal availability that met the enrollment criteria were difficult to source during the recruitment period. Extensive respiratory disease was prevalent in many potential animals that met the desired BCS (1 or 2). During the enrollment process an additional 12 horses were assessed for respiratory disease and all were positive EHV-1. Given the risk of transmission and morbidity rate for infectious respiratory diseases, alongside physiologic changes present during starvation, the pool of horses available to utilize for the project was limited [23].

Despite low sample numbers, power analysis prior to study approval and commencement indicated that animal numbers utilized in this study are adequate to detect statistical significance between treatment groups. Additionally, equine nutrition studies utilizing low animal numbers have been successfully completed and contribute meaningful information to the existing literature to advance clinical and management level decisions [24,25,26].

It is important to note that the backgrounds and histories of the horses enrolled in this study were largely unknown due to the variance in origin and ownership of the equine subjects. For this reason, it was not possible to consider factors that may influence the metabolome, such as diet history and the presence of past or current parasites, comorbidities, and disease. Perhaps these factors should not be considered as no two subjects will present with identical histories and such diversity contributes to a better overall understanding of rehabilitating emaciated horses. Despite these unknown variables, statistical analyses conducted on the refeeding process indicate a distinct difference between the equine metabolome during the CCP vs. the RP. This study supports the necessity of the refeeding process for the malnourished horse via blood chemistry and metabolomics analyses, despite its history and individual attributes.

### 4.2. Blood Chemistry Analysis Suggests Improved Biological Function during the RP

Statistical analysis of each subject’s blood chemistry during the refeeding process yielded significant changes in albumin, lactose dehydrogenase, aspartate amino transferase, creatinine, and non-esterified fatty acid levels between the CCP and RP. Albumin, a family of proteins primarily synthesized by the liver, was detected at lower levels during the CCP compared to the RP [27]. Low albumin levels in equine were previously thought to be associated with liver disease and/or liver failure, but a retrospective study of literature data concluded that consistently low albumin levels (hypoalbuminemia) reported were not directly associated with severe liver disease [27]. This finding does not dismiss the possibility that low albumin levels can be associated with improper hepatic function due to insufficient biological resources, such as those unobtainable during starvation.

Creatinine, a small molecule prominently present in plasma, was significantly altered between the CCP vs. RP. This nitrogenous molecule [28,29]. used to evaluate renal function and overall muscle mass was found to be lower during the CCP than that of the RP [30]. This compound is consistently produced in equine as a function of their muscle mass and body weight, dependent on proper renal function [30]. This finding directly correlates to increasing BCS scores during the refeeding process; as the malnourished horse undergoes the refeeding process and their BCS score increases, a positive correlation with increasing creatinine levels is observed, most likely due to a gain in muscle mass and body weight.

Levels of non-esterified fatty acids, compounds directly related to metabolic energy, were considerably lower during the RP. A lack of resources utilized for energy production (i.e., nutrient consumption) combined with consistent energy demands results in an overall negative energy balance [31]. Increased levels of non-esterified fatty acids have been positively correlated with an overall negative energy balance in many fasted mammals, including equine, as body fat is broken down (into non-esterified fatty acids) and utilized as an energy resource as a last resort [32,33]. The study of starved horses supports this finding as higher levels of these compounds were observed during a state of malnourishment (i.e., the CCP) and decreased during the RP, suggesting that the energy balance was restored once the horse was provided with proper nutrients.

Lactate dehydrogenase and aspartate aminotransferase are two enzymes that serve essential roles in anaerobic and aerobic metabolic pathways, respectively, with important emphasis in hepatic, renal, and skeletal muscular metabolism [33,34]. Interestingly, these enzymes serve as non-specific, clinical biomarkers of disease as their high-level presence is associated with cellular death [32,33]. In the present study, levels of these enzymes were statistically higher in equine during the RP than the CCP, which was an unanticipated observation as horses are not likely to experience as much cellular death during the RP compared to the CCP. Interestingly, increased levels of lactate dehydrogenase and aspartate aminotransferase have also been observed in a Japanese breed of equine that was fed a roughage-based diet; it was theorized that these enzymes were primarily higher due to diet [35]. Despite statistical increases observed in the present study, the average levels of these enzymes were within reference ranges for equine in both the CCP and RP, suggesting an increase not necessarily indicative of disease [36].

Blood chemistry panels are an efficient way to measure overall health and function of major organs, such as the liver and kidneys. However, these panels are limited in the amount of information attainable due to the sensitivity of methods and instrumentation. This problem of sensitivity is solved when characterizing blood (i.e., plasma) via metabolomics methods, which provide a vast array of quantitative information on the complexities of numerous metabolic pathways. Though metabolomics methods only measure small molecules (unlike blood chemistries which can detect small molecules, proteins, and enzymes), the variety of metabolites detected and low concentrations at which these compounds can be measured provide more global information on the metabolic processes, without the need for separate protein or enzyme analysis. Therefore, blood chemistries were used herein as a supplement to the more globally informative metabolomics analysis.

### 4.3. Potentially Toxic Metabolites and Indicators of Oxidative Stress Were in Lower Abundance during the RP

Fold change analysis of metabolites detected during the RP compared to the CCP showed perturbations in metabolites with varying biological activity (Table 5). It is important to note that fold change analyses of global metabolomics data may show increases or decreases in overall abundance but cannot discern the ultimate metabolic fate of a particular metabolite. For example, if a metabolite is found in higher abundance, it is possible that this compound is produced by the biological system at a higher rate, resulting in higher abundance; alternatively, this compound may be utilized by the system at a lower rate that also results in a higher concentration. To determine which of these scenarios is actually occurring, studies utilizing stable-isotope labeled metabolite tracers would be required to measure the fluxes of the metabolites of interest and ascertain the activity of pathways utilizing these molecules [37]. Nonetheless, one can often infer whether metabolites are being actively metabolized by looking at changes in several or all molecules in pathways in the context of the system state instead of analyzing data for individual molecules.

During the study, it was observed that metabolites shown to exhibit toxicity when found in high concentrations were found in lower abundance during the RP. In particular, allantoin, phenylalanine, and methionine were found to be significantly decreased in horses with higher BCS scores. Allantoin and methionine are both small molecules that are often considered indicators of oxidative stress, a condition marked by increased levels of free radical species that is often associated with disease and impaired nutritional status [43]. Specifically, methionine has been observed to be enhanced in subjects with severe acute malnutrition [43,44,45,46]. Because these potential biomarkers were found in lower abundance during the RP, it is theorized that the equine subjects are likely experiencing less oxidative stress due to these compounds acting as a defense against free radical species or their production at lower rates as their biological need to dissolve the radical species is lessened.

Similarly, phenylalanine appeared in higher abundance in equine during the CCP. Despite phenylalanine’s proteinogenic properties, it serves as a toxic metabolite when found at chronically high concentrations. As this compound is an essential amino acid, an overall reduction in this compound’s abundance during the RP may indicate an increase in health as this amino acid is potentially shunted to upregulate protein synthesis.

### 4.4. Metabolites Regulated by Skeletal Muscle Were Altered between the CCP and RP

Creatine was found to be significantly higher in abundance during the CCP. This finding supports a similar study conducted in juvenile pigs, which revealed that creatine levels were more abundant during a state of malnourishment [9]. This finding could be accredited to muscle wasting, a process in which muscle is broken down to be utilized for energy [47]. It is theorized that muscle wasting may cause the release of creatine from the muscle cells into circulation, yielding higher plasma creatine levels in horses with lower body condition scores. It has been observed that creatine supplementation does not significantly increase plasma creatine level, furthering the notion that increased plasma creatine abundance in horses during the CCP is due to muscle wasting [48]. As previously mentioned, methionine (a catalytic precursor of creatine) was also found in lower abundance in horses during the RP.

Glutamine is an important non-essential amino acid highly expressed in skeletal muscle, similar to that of creatine. Approximately 90% of all glutamine synthesized is via skeletal muscle; it is then transferred to the blood, where it represents the most abundant free amino acid [44]. Interestingly, this amino acid was present in lower abundance during the RP. Low levels of glutamine are often associated with disease, as consistent levels are required for proper function of many metabolic processes and organs (brain, lungs, gut, immune system, skeletal muscle, kidney, etc.). With this information and the observation that other toxic metabolites appear to decrease during the RP, it is more likely that glutamine levels are decreased in equine with higher BCS’s due to high utilization by the system for other biological processes.

In addition to creatine, two metabolites commonly associated with skeletal muscle (alanine and sarcosine) were detected. The methods of detection utilized in this study were unable to differentiate these two isomeric compounds, yet a notable feature was detected either representing the isomers together or separately. In the event this feature is significantly changed due to a decrease in sarcosine (a product of creatine), a simple explanation of decreased abundance of precursors during the RP may be offered. Though, an alternative hypothesis is that muscle wasting and oxidative stress were overall less prevalent during the RP, resulting in decreased sarcosine abundance. Regardless, this cascade of metabolites was important to the discernment of the CCP and RPs. However, the metabolic feature associated with this finding is more likely due to the presence of alanine, as this amino acid is one of the more highly abundant amino acids found in plasma and is known to be found in higher concentrations than sarcosine [49,50]. There are many theories as to why alanine was present in higher abundance during the CCP (including muscle wasting), but perhaps the most compelling is that of the shift in energy metabolism during starvation coordinated by metabolic crosstalk between skeletal muscle and the liver [51]. Gluconeogenesis, glycogenolysis, glycolysis, and protein breakdown all retain the common factor alanine transaminase, an enzyme that produces alanine partly to prevent skeletal muscle from accumulating toxic nitrogenous products and to produce hepatic glucose [51]. Thus, this offers an alternative to glucose homeostasis and energy metabolism [51]. Alanine levels have also been observed to increase in equine during and after exercise (i.e., when energy requirements change), which may mimic that of starvation [49]. Perhaps alanine was observed in greater abundance during the CCP due to high production amounts to satisfy the need of energy metabolism via hepatic pathways; yet, additional metabolites detected in this study provide evidence that hepatic function may be compromised, resulting in a build-up of this alternative energy product.

### 4.5. Significant Differences Were Observed in Metabolites Related to Liver and Kidney Function during the CCP and RPs

During the RP, uric acid, allantoin, and uridine were found in lower abundance while *N*-acetylornithine was found to be significantly increased. Uric acid is a purine derivative synthesized in the liver and, in excess amounts, is converted to allantoin, which is then filtered and excreted by the kidneys [52]. In mammals, allantoin reflects an endpoint of purine metabolism, unless it is shunted through the glyoxylate or glycine, serine, and threonine metabolism pathways [40,41,42]. The observation of altered levels of allantoin could be attributed to kidney function; starved, malnourished horses may experience kidney disfunction due to a lack of nutrients and resources, which prevent the ability of the kidneys to filter allantoin from the body. However, because uric acid is a precursor to allantoin and is not utilized in other metabolic processes (other than bile secretion), decreased levels of this compound indicate that it is being produced at a lower rate, yielding a lower rate of allantoin production [40,41,42]. Perhaps this is due to a lessened biological need for purine degradation, as these molecules are being directed to other pathways and are not present in excess.

Like uric acid and allantoin, uridine, a pyrimidine nucleotide important to RNA synthesis, is synthesized in the liver and excreted by the kidneys, if not catabolized [53]. Similar hypotheses exist for uridine as uric acid and allantoin. Lower levels are possibly detected during the RP as RNA is synthesized at a higher rate (i.e., less excess pyrimidine), which may indicate an increase in liver function. Yet, it’s also likely that uridine is better excreted by the kidneys during this phase, resulting in lower plasma levels.

The only metabolite found to be in higher abundance during the RP was that of *N*-acetylornithine, a small molecule involved in the biosynthesis of amino acids and a precursor to the urea cycle during arginine biosynthesis [40,41,42]. Endogenous arginine synthesis and the urea cycle primarily occur in the kidneys [54]. Because *N*-acetylornithine was the only significantly altered metabolite involved in this cascade (arginine, citrulline, and ornithine fold changes were not statistically different), increased production and subsequent utilization in the cycle is less likely than a build-up due to non-use. Isomers of *N*-acetylornithine have been detected in high concentrations in various food sources, such as wheats and some fruits and vegetables. Thus, it is more likely that horses with higher BCSs that have undergone the refeeding process accumulate higher levels of *N*-acetylornithine in plasma from their diet.

### 4.6. The Plasma Metabolome Illuminates Differences in Equine Subject-Specific Attributes

Statistically significant differences between specific metabolites and metabolic profiles were observed between the CCP and RP for ten equine subjects of various sex, age, breed, and origin. Differences were evident between designated diet, mares and geldings, as well as age group (below 20 years old and above 20 years old); yet, these differences were not evident in the analysis of the CCP vs. RP (e.g., sex-specific clusters were not evident in the CCP vs. RP discriminant analysis), indicating that these subject-specific attributes did not affect the conclusion that the metabolomic profiles of horses undergoing the refeeding process were statistically different.

As diet played an essential role in the current study, an analysis to determine potential differences in metabolomic profiles in equine assigned to a particular diet was conducted. Various statistical analyses suggest perturbations in metabolism with respect to diet. In the event of a negative energy balance (e.g., in the case of malnourishment), particular care is placed on diet such that the subject can consume and digest nutrients without inducing refeeding syndrome or overfeeding (i.e., the process of introducing nutrients in excess such that metabolic homeostasis cannot be maintained [55]. Overfeeding may result in further metabolic damage, particularly in the liver and kidneys, both major organs which have shown to be impaired during the CCP in the current study. Thus, diets for malnourished equine must be carefully tailored to slowly reintroduce nutrients in a manner such that further metabolic damage is not inflicted. Though an in-depth analysis of the effects of a particular diet or set of nutrients on equine outcome was not within the scope of this study, it is worth noting the differences observed in the global metabolome in this study; further investigation is required to determine the detailed effects of diet and nutrients that cause this separation.

With advances in equine feed that cater to aging horses, the average lifespan of the equid is increasing, which has prompted a surge in research devoted to older horses (i.e., horses 20 years old and above) [56]. For this reason, an analysis to evaluate the metabolome of younger vs. older horses in the present study was conducted and resulted in clear differences between the two groups. The amount of metabolites that drove this distinction was not as prominent as other comparisons (i.e., diet and sex), though various statistical tests support perturbations based on age group. It is theorized that many metabolic factors may play a role in this age-based separation, including alterations in muscle fiber type and renal function [56]. Similar to effects from diet, further studies are warranted to establish clear relationships between the aging horse and metabolism.

It is important to note that the analysis of the metabolome provides a biochemical snapshot of the metabolic state for the system, which is highly complex. For this reason, it is not surprising that attributes such as diet, sex, and age affect metabolic profiles. Moreover, these attributes contribute diversity to the study of the impacts of refeeding to gain a better understanding of caring for malnourished equine.

## 5. Conclusions

Comparison of the equine metabolome during a state of malnourishment versus rehabilitation illuminated significant differences that provide essential biological insight to the process of refeeding the emaciated horse. After commencement of the refeeding process (i.e., during the RP), a decrease in potentially toxic metabolites was observed, metabolic alterations suggesting increased liver and kidney function were shown, and metabolite changes related to muscle gain/repair and changing energy demands were evident. Blood chemistry panels also showed statistical differences between analyte abundance between the CCP (i.e., state of malnourishment) and the RP, further indicating improper liver and kidney function during the CCP. The combination of blood chemistry and metabolomics analyses on the equine subjects in this study provided essential insight to the biological processes that are influenced when rehabilitating the malnourished horse. Despite differences observed between the metabolome with respect to subject-specific attributes (i.e., diet, sex, and age group), a clear discernment between the metabolome unique to the CCP and RP were observed, providing evidence that the refeeding process has a profound impact on equine rehabilitation.

## Figures and Tables

**Figure 1 animals-12-02527-f001:**
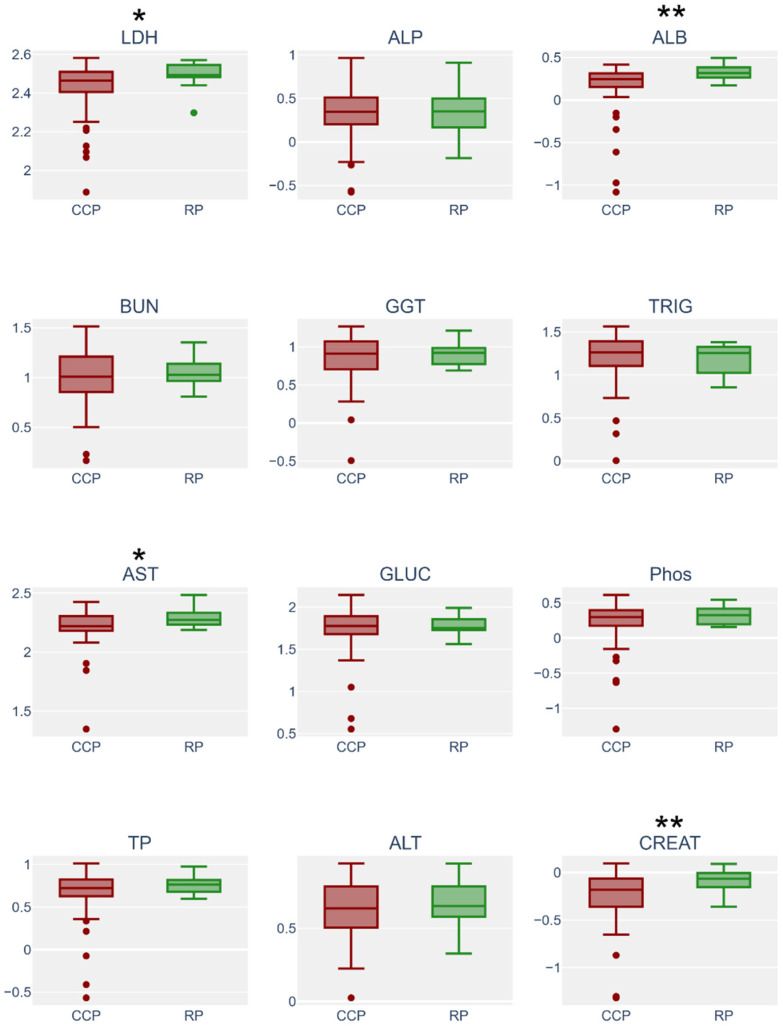
Box plots of blood plasma data conveying the differences between blood during the CCP (*n* = 68) and the RP (*n* = 15). Asterisks represent statistical significance (*p* value ≤ 0.1 = *; *p* value ≤ 0.05 = **). LDH: lactate dehydrogenase (*p* value 0.06), U/L; ALP: alkaline phosphatase, U/L; ALB: albumin (*p* value 0.04), g/dL; BUN: blood urea nitrogen, mg/dL; GGT: gamma-glutamyl transferase, U/L; mg/dL; TRIG: triglycerides, mg/dL; AST: aspartate amino transferase (*p* value 0.09), U/L; TP: total protein, g/dL; GLUC: glucose, mg/dL; Phos: phosphate, mg/dL; ALT: alanine transaminase, U/L; CREAT: creatinine (*p* value 0.04).

**Figure 2 animals-12-02527-f002:**
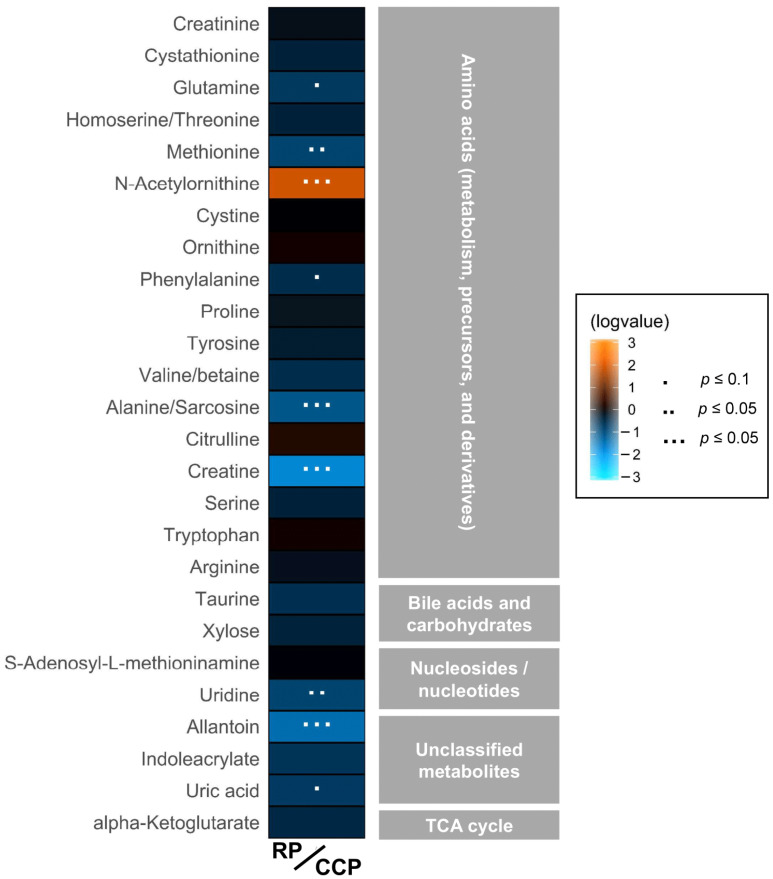
Heatmap showing increased or decreased abundance of detected metabolites organized by compound class between the RP (*n* = 20) and the CCP (*n* = 68; fold change of RP/CCP). Eight compounds exhibited significantly lower abundance while one compound showed significantly higher abundance during the RP.

**Figure 3 animals-12-02527-f003:**
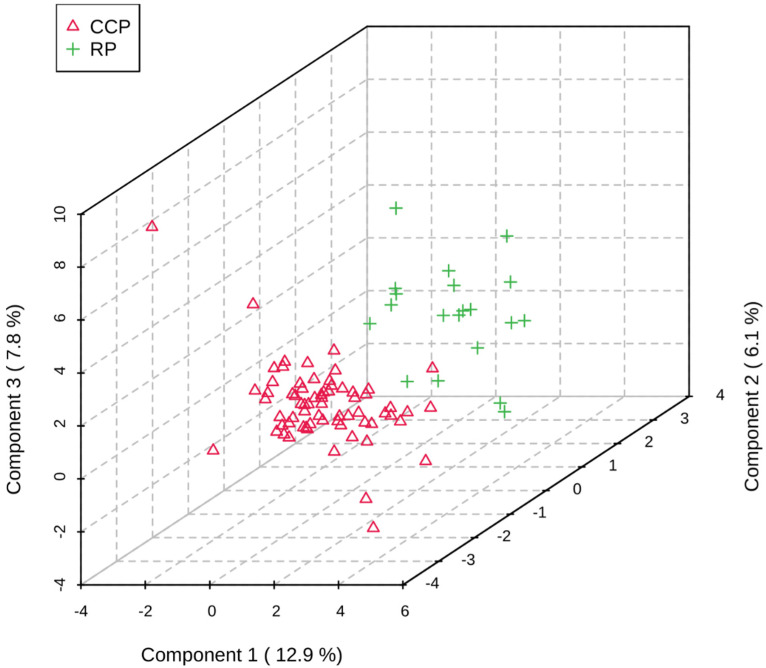
PLS-DA containing known metabolites and unidentified features detected during the CCP (*n* = 68) and RP (*n* = 20). The resulting plot shows separation between the groupings, indicating differences in the equine metabolomic profile during the CCP and the RP.

**Figure 4 animals-12-02527-f004:**
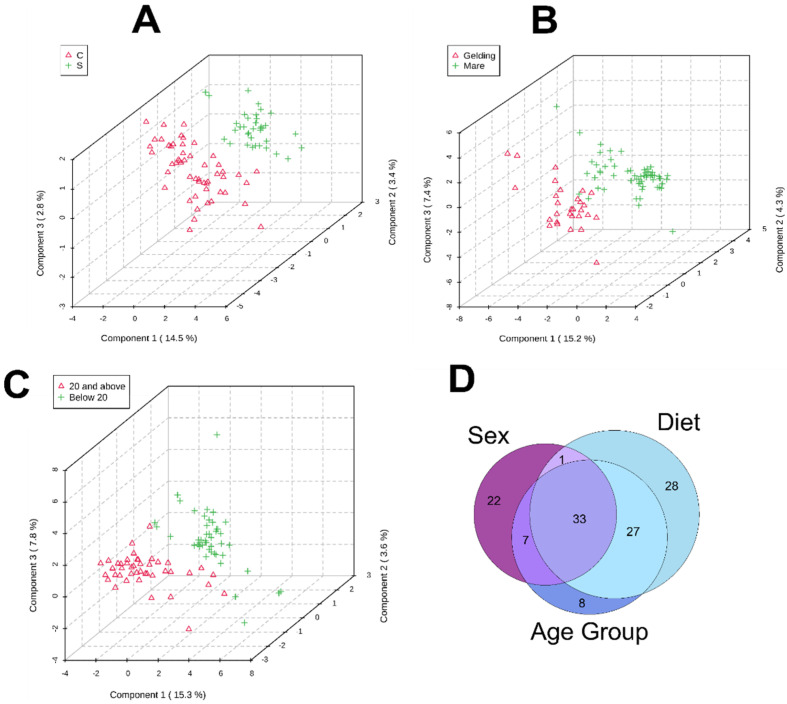
PLS-DAs of known metabolites and unidentified features detected in the metabolomes of all horses that were enrolled in the refeeding program grouped by (**A**) diet (C vs. S), (**B**) sex (mare vs. gelding), and (**C**) age group (below 20 and 20 and above). Sample sizes used for PLS-DAs were as follows C: *n* = 52; S: *n* = 36; mare: *n* = 59; gelding: *n* = 29; below 20: *n* = 45; 20 and above: *n* = 43. Venn Diagram (**D**) displaying similarities and differences between features in each grouping.

**Table 1 animals-12-02527-t001:** Available metadata for equine subjects enrolled in the refeeding program.

Subject Identifier	Sex	Breed	Approximate Age (Years)
A	Mare	Quarter Horse	19
A2	Mare	Tennessee Walking Horse	22
B	Mare	Appendix	20
C	Mare	Quarter Horse	20
D	Mare	Tennessee Walking Horse	20
E	Mare	Quarter Horse/Arabian	18
F	Gelding	Appendix	16
G	Gelding	Appendix	23
H	Mare	Standardbred	15
I	Gelding	Quarter Horse	17

**Table 2 animals-12-02527-t002:** Nutrient analysis of forage of diet components.

Nutrient	Timothy Hay	Equine Senior	Free Balance Mineral
Digestible Energy (mCal/kg)	1.95	2.7	0
Protein (%)	9.88	15.5	1.55
Lysine (%)	0.34	0.7	N/A
Fat (%)	1.29	6.2	0.1
Acid Detergent Fiber (%)	39.50	22.54	N/A
Neutral Detergent Fiber (%)	62.23	38.89	N/A
Calcium (%)	0.21	0.77	13.52
Phosphate (%)	0.27	0.55	12.66
Potassium (%)	2.39	1.6	0.66

**Table 3 animals-12-02527-t003:** Measured parameters for blood plasma and serum.

Analyte	Abbreviation	Units	Matrix
Lactate dehydrogenase	LDH	U/L	Plasma/serum
Alkaline phosphatase	ALP	U/L	Plasma/serum
Albumin	ALB	g/dL	Plasma/serum
Blood urea nitrogen	BUN	mg/dL	Plasma/serum
Gamma-glutamyl transferase	GGT	U/L	Plasma/serum
Aspartate amino transferase	AST	U/L	Plasma/serum
Glucose	GLUC	mg/dL	Plasma/serum
Phosphate	PHOS	mg/dL	Plasma/serum
Total protein	TP	g/dL	Plasma/serum
Alanine transaminase	ALT	U/L	Plasma/serum
Creatinine	CREAT	mg/dL	Plasma/serum
Triglycerides	TRIG	mg/dL	Plasma/serum
Creatine kinase	CK	U/L	Serum only
Non-esterified fatty acids	NEFA	mEq/L	Serum only
Serum calcium	SCA	mg/dL	Serum only

**Table 4 animals-12-02527-t004:** Outcomes for equine enrolled in the refeeding program.

Subject Identifier	Program Outcome	Samples Taken	
		CCP	RP
A	Did not complete	5	NA
A2	Did not complete	7	NA
B	Did not complete	7	NA
C	Completed	7	4
D	Did not complete	7	1
E	Completed	7	4
F	Completed	7	3
G	Completed	7	3
H	Completed	7	3
I	Completed	7	2

NA: not applicable.

**Table 5 animals-12-02527-t005:** Biological significance of metabolites observed to have significantly increased or decreased between the RP and the CCP.

Metabolite Class	Significantly Increased or Decreased (RP/CCP)	Metabolite	Biological Significance ^1^
Amino acids (metabolism, precursors, derivatives)	Decreased	Glutamine	Important non-essential amino acid Synthesized by skeletal muscle
	Decreased	Methionine	Sulfur-containing amino acid indicative of oxidative stress at high concentrations
	Increased	*N*-acetylornithine	Primary metabolite commonly found in food sources (e.g., wheat) Precursor to the urea cycle
	Decreased	Phenylalanine	Essential amino acid that acts as a toxin at high concentrations Used in the synthesis of proteins
	Decreased ^2^	Alanine	Highly concentrated in muscle Functions as a major energy source and regulator of glucose metabolism
		Sarcosine	Often detected in muscle Metabolic product of creatine
	Decreased	Creatine	Generally found in skeletal muscle Needs *methionine* as a catalytic precursor Metabolizes to energy products, urea, or sarcosine
Nucleosides/nucleotides	Decreased	Uridine	High levels associated with diet supplementation
Unclassified metabolites	Decreased	Uric acid	Final product of purine metabolism Metabolic precursor to *allantoin* Potentially a toxic metabolite and is often excreted
	Decreased	Allantoin	Metabolic byproduct of *uric acid* Indicator of oxidative stress

^1^ Biological significance gathered from the Human Metabolome database (https://hmdb.ca/ (accessed on 31 July 2022); [38]), the Livestock Metabolome database (https://lmdb.ca/ (accessed on 31 July 2022); [39]), and Kyoto Encyclopedia of Genes and Genomes (KEGG) database (https://www.genome.jp/kegg/ (accessed on 31 July 2022); [40,41,42]). ^2^ These metabolites cannot be differentiated via the employed method; therefore, it is unknown whether they both decrease or singly decrease.

## Data Availability

The metabolomics data has been submitted to the MetaboLights database under study MTBLS5566 (www.ebi.ac.uk/metabolights/MTBLS5566).

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
