# Peer review of "Metabolomic Profiles in Starved Light Breed Horses during the Refeeding Process"

_animals, 2022, doi:10.3390/ani12192527_

Round 1

Reviewer 1 Report

The manuscript reports investigations of the metabolome of malnourished horses undergoing a refeeding process.

Major comments:

In my opinion, the topic of the manuscript is to closely restricted. The described study design and sampling contained samples at the starting point so the initial status of the horses can be determined. Also final samples were taken before the horses were disenrolled from the study. Samples of starting and end conditions of the animals should be included into the manuscript to better characterize the effects of the refeeding process on the horse metabolome and to improve the information of the manuscript.

In general, the number of horses enrolled in the study is very low. And it was reduced during the study period because of infections of 4 horses. At the end only 6 horses are completely included. It is understandable, that the feeding and housing of horses is much more expensive than for smaller animals. But the low number of group members must be taken into account for the statistical analysis. For t-tests and ANOVA (and also for multivariate methods like PLS-DA) data have to be normally distributed. This assumption is not given for data from very small groups (n = 3 – 10). That’s why the use of t-test and ANOVA to analyze these data is problematic. Did you test your data for normal distribution? Other methods (like Mann-Whitney-U-Test or Kruskal-Wallis-Test) can be used as nonparametric alternatives. Furthermore, it makes no sense to statistically investigate groups with only one or two members (e.g. breed).

Because of the small number of horses it would be very helpful to know which horses are excluded from the study.

You investigated the changes in the horse blood chemistry and metabolome by comparison of RP to CCP. You described in the manuscript, that daily samples were taken during CCP (10 samples/animal). How many samples were taken during RP? For me it is not clear, which samples are included in the analysis for comparing RP with CCP. Did you use the mean (or median) of all samples (all sampling time points) from CCP/RP? Differences of values in RP between BSCs are only shown in Figure 3.

The study design included two diet groups for the horses, but the assignment of the horses to the groups was not described. Additionally, no effects of the different diets on the metabolome of the animals were described in the manuscript.

Minor comments:

It is not clear, how many horses and sampling time points are included in the data analysis and how many horses are members of the groups used for t-test or ANOVA. Please add this information for each figure and table.

It would be also interesting to know, how much metabolites were identified and how much unknown features are included. Could you please attach a list of the identified metabolites?

The fold change analysis of the plasma metabolites is mentioned in chapter 3.3. Could you please attach tables containing the calculated values for all group comparisons?

It is difficult to compare data from plasma or other body fluids. Did you use any internal standard to compare the quality of the extraction procedure? Did you use any kind of data normalization, scaling or transformation?

For the metabolome data you did a correction for multiple comparisons using FDR. Did you do the same for the blood chemistry analytes?

line 78: A lot of metabolome investigations were done for horses but mostly dealing with endurance exercise, doping analyses, horse specific diseases.

line 129: What was the starting BSC of the horses? How many blood samples were taken from each horse during RP?

line 135: What is the meaning of “horses had final samples taken”?

line 166: Why you used this extraction procedure (described for E. coli) for the plasma samples?

line 247: Please correct the chemical name N-acetylorninthine.

line 316: What is the meaning of the age groups? Is there any important change in horse physiology at the age of 20? What do you expect as the outcome comparing these two age groups?

line 327: Please delete.

Table 4: Please replace “Creatine either produces energy products, urea, or sarcosine”, “Uric Acid…Produces allantoin, and “Allantoin… Produced by uric acid”. They can be metabolized or can react to …

line 424: s is missing – stable-isotope labeled

line 445: Phenylalanine is an essential amino acid (also for horses) and cannot be produced by the horse, only taken up or released by protein degradation.

Author Response

Response to reviewers’ comments

Reviewer 1 

The manuscript reports investigations of the metabolome of malnourished horses undergoing a refeeding process. 

Major comments: 

In my opinion, the topic of the manuscript is to closely restricted. The described study design and sampling contained samples at the starting point so the initial status of the horses can be determined. Also final samples were taken before the horses were disenrolled from the study. Samples of starting and end conditions of the animals should be included into the manuscript to better characterize the effects of the refeeding process on the horse metabolome and to improve the information of the manuscript. 

Response: After addressing the comments below, we hope that the study design, sampling, and data are now more clearly presented. 

In general, the number of horses enrolled in the study is very low. And it was reduced during the study period because of infections of 4 horses. At the end only 6 horses are completely included. It is understandable, that the feeding and housing of horses is much more expensive than for smaller animals. But the low number of group members must be taken into account for the statistical analysis. For t-tests and ANOVA (and also for multivariate methods like PLS-DA) data have to be normally distributed. This assumption is not given for data from very small groups (n = 3 – 10). That’s why the use of t-test and ANOVA to analyze these data is problematic. Did you test your data for normal distribution? Other methods (like Mann-Whitney-U-Test or Kruskal-Wallis-Test) can be used as nonparametric alternatives. Furthermore, it makes no sense to statistically investigate groups with only one or two members (e.g. breed). 

Response: A power analysis was performed prior to IACUC approval and study commencement to ensure that despite low animal numbers, statistical differences could be detected. Power analysis was completed through statistical consultation with a resident statistician to ensure all variables were considered. References to other published nutrition-based studies in horses and other equids have been added to illustrate that despite low animal numbers, impactful results from a statistical and physiological perspective can be achieved. Please reference Section 4.1 (line 423) for these additions.  

Taking the latter portion of this comment into consideration, we have removed breed analyses via ANOVA and ensured that all other comparisons contained multiple replicates (i.e., diet, sex, and age group). We also ensured data were normally distributed after normalization and log transformation. We hope the re-evaluation of this data addresses concerns with the data analysis. Please reference Figure 4 (line 406), Section 3.5 (line 350), and Section 4.6 (line 638) for these changes. 

Because of the small number of horses it would be very helpful to know which horses are excluded from the study. 

Response: Table 4 (line 248) has been added to clarify which horses completed/did not complete the program as well as how many samples were taken from each horse during the program. 

You investigated the changes in the horse blood chemistry and metabolome by comparison of RP to CCP. You described in the manuscript, that daily samples were taken during CCP (10 samples/animal). How many samples were taken during RP? For me it is not clear, which samples are included in the analysis for comparing RP with CCP. Did you use the mean (or median) of all samples (all sampling time points) from CCP/RP? Differences of values in RP between BSCs are only shown in Figure 3. 

Response: We hope that this question is now answered with the addition of Table 4 (line 248).  

The study design included two diet groups for the horses, but the assignment of the horses to the groups was not described. Additionally, no effects of the different diets on the metabolome of the animals were described in the manuscript. 

Response: Thank you for bringing up this point – this was an oversight that needed to be addressed. The manuscript has been reformatted to include analysis of diet (similar to how we evaluated differences in sex and age group, now in Figure 4). We believe that the effects of the diet on the metabolome warrant further investigation and simply call out our general observations in this study. We refer you to the following sections for these changes: Figure 4 (line 406), Section 3.5 (line 350), and Section 4.6 (line 638). 

Minor comments: 

It is not clear, how many horses and sampling time points are included in the data analysis and how many horses are members of the groups used for t-test or ANOVA. Please add this information for each figure and table. 

Response: All figures showing data now have replicate information added for clarity. 

It would be also interesting to know, how much metabolites were identified and how much unknown features are included. Could you please attach a list of the identified metabolites? 

Response: This information has now been added into the Data Analysis section of the Methods (line 214). Identified metabolites are listed in Figure 2 (line 297).  

The fold change analysis of the plasma metabolites is mentioned in chapter 3.3. Could you please attach tables containing the calculated values for all group comparisons? 

Response: Yes, absolutely. T-test information was included in the supplementary information, but not called out in the manuscript – our apologies for the oversight. Fold change analysis and t-test information have now been called out in the manuscript text (line 294), but is provided in the supplementary information for brevity.  

It is difficult to compare data from plasma or other body fluids. Did you use any internal standard to compare the quality of the extraction procedure? Did you use any kind of data normalization, scaling or transformation? 

Response: Thank you for bringing this topic to the forefront. Blood chemistry data were not normalized in the same manner as metabolites – we agree this was not the ideal method of analysis. Data were re-evaluated (i.e., metabolomics data and blood chemistry data were normalized and transformed in the same manner) such that they are comparable. This did change some results (e.g., triglycerides no longer showed statistical changes), but the overall conclusions remain the same. Methods (line 224), results (Section 3.2; line 262), and discussion sections (Section 4.2; line 451) have been edited per this comment and the re-evaluation of data. 

For the metabolome data you did a correction for multiple comparisons using FDR. Did you do the same for the blood chemistry analytes? 

Response: To clarify, Metaboanalyst allows FDR cutoffs, which is what we initially used. For clarity and consistency, during data re-evaluation, we used a p value cutoff of 0.1 for all analyses (specifically, nothing above 0.1 was reported as significant by Metaboanalyst). 

line 78: A lot of metabolome investigations were done for horses but mostly dealing with endurance exercise, doping analyses, horse specific diseases. 

Response: We have provided clarification to the current research on equine metabolomics (line 80). 

line 129: What was the starting BSC of the horses? How many blood samples were taken from each horse during RP? 

Response: All horses showed an initial BCS of 2 or less. We originally indicated this in the methods (line 100), but have also added this to the results (line 233) for clarity. The number of samples taken during the CCP and RP are now indicated in Table 4 (line 248). 

line 135: What is the meaning of “horses had final samples taken”? 

Response: Clarification added to elaborate on final data collection prior to study disenrollment (line 137).  

line 166: Why you used this extraction procedure (described for E. coli) for the plasma samples? 

Response: Here at the Biological Small Molecule Mass Spectrometry Core at the University of Tennessee Knoxville, we adopt the extraction procedure from Dr. Joshua Rabinowitz, which has shown to be applicable for various biological matrices, including plasma. Some references are provided below in which the procedure we utilized in this manuscript was applied to other biological matrices (including plasma, serum, tissue, etc.).  

  • Denny, J.E., et al. Scientific Reports. 2019, 9(3472) https://doi.org/10.1038/s41598-019-40266-6
  • Clemmons, B.A., et al. Scientific Reports. 2019, 9(1) doi: 10.1038/s41598-019-55978-y.
  • Tague, E.D., et al. Journal of Medicinal Food. 2018, 21(3) https://doi.org/10.1089/jmf.2017.0062

line 247: Please correct the chemical name N-acetylorninthine. 

Response: All mentions of this compound have been checked and corrected if necessary. 

line 316: What is the meaning of the age groups? Is there any important change in horse physiology at the age of 20? What do you expect as the outcome comparing these two age groups? 

Response: There is an interesting review (cited below) that defines “older horses” aged 20 and above and cites studies that focus on the muscle composition, renal function, etc. of aged horses. Thus, we investigated whether or not there were differences between age groups (below 20 vs. 20 and above) in our study, which does support metabolic differences between these age groups. 

McKeever, K. H. Clinical Techniques in Equine Practice. 2003, 2 (3), 258-265. https://doi.org/10.1053/S1534-7516(03)00068-4

line 327: Please delete. 

Response: The line “Figures, Tables, and Schemes” has been deleted. 

Table 4: Please replace “Creatine either produces energy products, urea, or sarcosine”, “Uric Acid…Produces allantoin, and “Allantoin… Produced by uric acid”. They can be metabolized or can react to … 

Response: All mentions of connected metabolites in Table 4 have been edited to language such as “...is metabolized to...” instead of “...produced by...”.  

line 424: s is missing – stable-isotope labeled 

Response: Thank you for catching this - “stable-isotope labeled” has been added. 

line 445: Phenylalanine is an essential amino acid (also for horses) and cannot be produced by the horse, only taken up or released by protein degradation. 

Response: The mention of phenylalanine being an essential amino acid has been added for clarification and the conclusion that it is produced at lower amounts has been removed (line 554) 

Reviewer 2 Report

Dear Authors,

The manuscript entitled" Metabolomic profiles in starved light breed horses during the refeeding period is interesting and has some good results. However some points need to be addressed before publication.

Is there any history on feeding when horses were enrolled in the study?

In the results you compare the two period of feeding. What about diet effects? Since you described both diets, a comparison should be available.

Discussion, line 529: should be 6 horses, not 10. What data wat utilized for the analyzing of the data?

Author Response

Reviewer 2 

Dear Authors, 

The manuscript entitled" Metabolomic profiles in starved light breed horses during the refeeding period is interesting and has some good results. However some points need to be addressed before publication. 

Is there any history on feeding when horses were enrolled in the study? 

Response: This information was included after the original manuscript section 4.5 in the discussion (now section 4.6). To increase clarity, information on lack of diet history and animal sourcing has been added under section 4.1 (lines 423) in the discussion.  

In the results you compare the two period of feeding. What about diet effects? Since you described both diets, a comparison should be available. 

Response: Thank you for bringing up this point – this was an oversight that needed to be addressed. The manuscript has been reformatted to include analysis of diet (similar to how we evaluated differences in sex and age group, now in Figure 4). We believe that the effects of the diet on the metabolome warrant further investigation and simply call out our general observations in this study. We refer you to the following sections for these changes: Figure 4 (line 406), Section 3.5 (line 350), and Section 4.6 (line 638).   

Discussion, line 529: should be 6 horses, not 10. What data was utilized for the analyzing of the data? 

Response: As health did not play a factor in these hypothesis generating comparisons, we did in fact utilize data from all ten horses for this part of the study. To clarify this point in the manuscript, we added Table 4 (line 248), which lists the completion/removal of each horse and how many samples were collected during the specified time periods. In the section you are referring to, we utilized metabolomics data – this has now been clarified in the text. 

Reviewer 3 Report

This is an important tribute to the equine veterinary medicine

Author Response

Reviewer 3 

This is an important tribute to the equine veterinary medicine 

Response: Thank you – we believe that readers of this special issue will agree and will aid in future management decisions for emaciated equids. 

Round 2

Reviewer 1 Report

Thank you very much for accepting my questions, your detailed answers and the correction of errors in your manuscript.